# Fast Charging of a Thermal Accumulator Based on Paraffin with the Addition of 0.3 wt. % rGO

Sergey A. Baskakov [1,2], Yulia V. Baskakova [1,2], Eugene N. Kabachkov [1,3,*], Elizaveta V. Dvoretskaya [1], Victor N. Vasilets [4], Zhi Li [5] and Yury M. Shulga [1,*]

1 Federal Research Center of Problems of Chemical Physics and Medicinal Chemistry, Russian Academy of Sciences, Academician Semenov Avenue 1, 142432 Chernogolovka, Moscow Region, Russia; baskakov@icp.ac.ru (S.A.B.)
2 GRAPHENOX LLC, Academician Semenov Avenue 1, Build 1/6, Room 6, 142432 Chernogolovka, Moscow Region, Russia
3 Institute of Solid State Physics, Russian Academy of Sciences, 2 Academician Osipyan str., 142432 Chernogolovka, Moscow Region, Russia
4 N.N. Semenov Federal Research Center for Chemical Physics, Russian Academy of Sciences (Branch), Academician Semenov Avenue 1/10, 142432 Chernogolovka, Moscow Region, Russia
5 School of Resources and Safety Engineering, Central South University, Changsha 410083, China
* Correspondence: en.kabachkov@gmail.com (E.N.K.); yshulga@gmail.com (Y.M.S.)

**Abstract:** The composite of paraffin with reduced graphene oxide (paraffin/rGO) was obtained at 70 °C by the mechanical mixing of the components followed by ultrasonic dispersion. The introduction of only 0.3 wt.% rGO stained the paraffin black. It has been shown that thermal batteries made from 160 grams of pure paraffin and 160 grams of the composite are equally slow at charging when placed in boiling water. However, two minutes of microwave heating increases the temperature of the pure paraffin battery to only 32 °C, while the temperature of the paraffin/rGO composite battery rises to 74 °C, which is above the paraffin solid–liquid phase transition temperature.

**Keywords:** reduced graphene oxide; paraffin; composites; phase change materials; thermal accumulator; microwave irradiation

## 1. Introduction

The collection and storage of thermal energy is currently an important problem, judging by the number of relevant publications (see reviews [1–5] and references to them). Among the materials that have been proposed to solve this problem, a special place is occupied by organic materials with a solid–liquid phase transition, which can store and release a large amount of energy during an isothermal phase transition. However, these materials have a number of significant drawbacks, including low thermal conductivity and fluid leakage.

A large number of works have been devoted to the elimination of fluid leakage. Publications [6–10] can be added to the already mentioned reviews. In our work, we do not touch on this important task. We are more concerned instead with the problem of the fast charging of heat accumulators, which is naturally related to the thermal conductivity of working material, which for such a typical organic phase changes materials (PCMs) as paraffin is only 0.14–0.3 W·m$^{-1}$·K$^{-1}$ [11].

Specific methods for controlling the thermal conductivity of organic materials with a solid–liquid phase transition have been described quite a lot. At the same time, it is common for all specific methods that materials with a high thermal conductivity are introduced into materials with a phase transition. Obviously, the latent heat of the composite is reduced compared to that of pure PCM. This implies an obvious conclusion—the concentration of additives with high thermal conductivity should be low. The distribution of the additive also has an important effect on the thermal conductivity of the composite [12].

The uniformity of the distribution of unbound particles of the additive can be significantly disturbed during phase cycling if the specific gravity of the additive differs significantly from the specific gravity of PCM. In this sense, carbon materials, especially GO, whose thermal conductivity is approximately 5000 W·m$^{-1}$·K$^{-1}$ [11], are extremely promising.

We note here that graphene, graphene oxide, and reduced graphene oxide appear to be promising materials for numerous interesting applications [13–22]. In principle, PCM composites with GO and rGO have already been described [23–29]. We do not distinguish between GO and rGO because there is currently no agreed boundary criterion for when a GO transitions into an rGO. Instead, it is assumed that GO is rGO with a low degree of reduction. We quote here only some of the indicated works. Let us immediately note that the obtained results unambiguously indicate that the addition of rGO increases the thermal conductivity of the composite. This is no longer necessary to prove. However, the remaining properties of the composites, despite the apparent simplicity of the system and standard research methods, do not agree with each other in all details, even in the case of a paraffin/rGO composite.

In 2013, a vacuum impregnation method was proposed to obtain a composite. The study showed that the composite obtained by this method with a low degree of reduction and a high rGO content (52 wt %) had a conductivity of 0.985 W·m$^{-1}$·K$^{-1}$ [23]. It was shown in [14] that the PCM composite with GO (0.3 wt %) was more thermally stable than the initial paraffin. At the same time, an increase in thermal conductivity and a decrease in the melting and solidification temperatures of the composite were also observed compared to pure PCM.

It is also known from the literature that GO can be efficiently heated by microwave radiation. This feature can be used for the exfoliation of GO [30,31]. rGO also exhibits an excellent microwave absorbing property [32–34].

The purpose of this work was to prepare and study the properties of a paraffin/rGO composite (0.3 wt %). The mass fraction of 0.3% was chosen based on earlier studies [24,25]. At the same time, we did not intend to clarify the data obtained earlier but to show how you could quickly charge a battery with the help of a large (160 g of paraffin) battery based on this composition. Note that we did not find publications in the literature that described the approach in this paper.

## 2. Materials and Methods

### 2.1. Materials

We used paraffin grade P-2 (GOST 23683-89), produced by AO REAKHIM LLC (Staraya Kupavna, Russia). Paraffin P-2 is highly purified paraffin, used for coating and the impregnation of flexible food packaging that retains its elasticity at low temperatures, as well as the components of alloys for coating wooden, concrete, metal containers intended for food storage, in the production of various wax formulations, medical equipment and cosmetics.

Reduced graphene oxide was obtained by the microwave exfoliation of graphene oxide powder according to the procedure described in [26].

### 2.2. Obtaining a Composite Paraffin/rGO

A total of 200 g of paraffin was placed in a beaker and heated in an oven at 70 °C until paraffin was completely melted. rGO powder was introduced into the molten paraffin at an amount of 0.3 wt.% of the paraffin mass and was mixed. rGO was then dispersed using a MELFIZ MF 93.1 ultrasonic disperser at a frequency of 22 kHz and a power of 600 W for 10 min. In the process of ultrasonic irradiation, the mixture was heated, which maintained the paraffin in a molten state. The resulting suspension of paraffin/rGO was poured into a heat accumulator case made of high-pressure polyethylene and hermetically sealed with a lid. A heat accumulator based on paraffin without the addition of rGO was prepared in a similar way.

*2.3. Equipment*

The IR spectra (resolution 1 cm$^{-1}$, number of scans 32) were recorded at room temperature in the range of 450–4000 cm$^{-1}$ on a Perkin-Elmer "Spectrum Two" Fourier-transform IR spectrometer (Waltham, MA, USA) with an ATR attachment.

The Raman spectra were obtained on a Bruker Senterra micro-Raman instrument (Bruker Optics GmbH, Ettlingen, Germany). The laser excitation wavelength was 532 nm, the power at the measurement point was 0.1 mW; the beam diameter was ~2 μm.

The thermogravimetric analysis (TG) of the samples was performed using an STA 449 F3 Jupiter instrument (Selb, Bavaria, Germany). To calibrate the balance, the chamber of the instrument was evacuated (10$^{-2}$ bar) and filled with He gas grade 6.0 (99.9999%). After that, two empty corundum (Al$_2$O$_3$) crucibles were placed on the holder in the working chamber of the device, and the baseline was recorded. Then, a sample was placed in one of the empty crucibles; the instrument chamber was again evacuated and filled with helium. The measurements were carried out in the temperature range of 20–550 °C at a rate of 10 °C/min and in a He flow of 50 ml/min.

DSC curves were recorded on a DSC 822 instrument (Mettler-Toledo, Bilbao, Spain). Samples weighing 10 mg were placed in an aluminum ampoule, which during the measurement, was in a nitrogen atmosphere at a flow rate of 50 ml/min. Heat release was measured in the temperature range from 0 to +90 °C at a heating rate of 5 °C/min. Melting and solidification temperatures were calculated using Mettler-Toledo software.

Thermograms were obtained using a Seek Thermal Compact portable thermal imager (Santa Barbara, CA, USA) connected to an Android smartphone. The viewing angle was 36 degrees. The device had a matrix of 206 by 156 pixels; the range of recorded temperatures was from −40 °C to 330 °C.

## 3. Results

*3.1. Photo of the Cuts*

Figure 1 shows a photograph of the sections of samples in the paraffin and composite aligned along the cut plane. It can be seen that the size of the white islands in the case of pure paraffin was larger than in the case of the composite. If we assume that the white islands were due to reflections of light from the mirror areas, then the question arises regarding their structure. It is possible that the reflections are related to the plane-parallel stacking of saturated hydrocarbon chains.

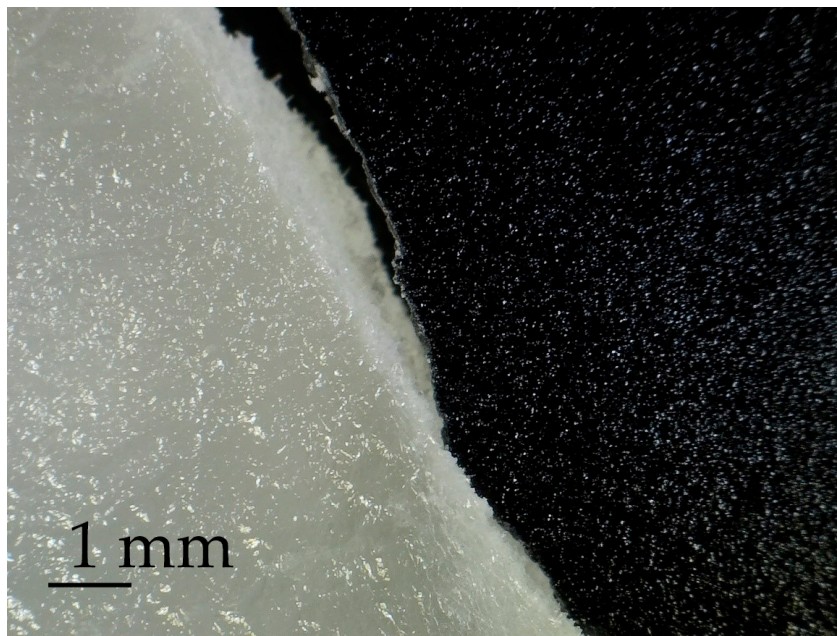

**Figure 1.** Optical photograph of a section (from left to right) of the original paraffin and composite.

### 3.2. IR Spectra

There are actually only two absorption bands in the IR spectrum of rGO (Figure 2): one at 1557 cm$^{-1}$ and the other (wide) in the range of 1230–1000 cm$^{-1}$ with a local extremum at 1160 cm$^{-1}$, which are due to C=C and C-OH [35].

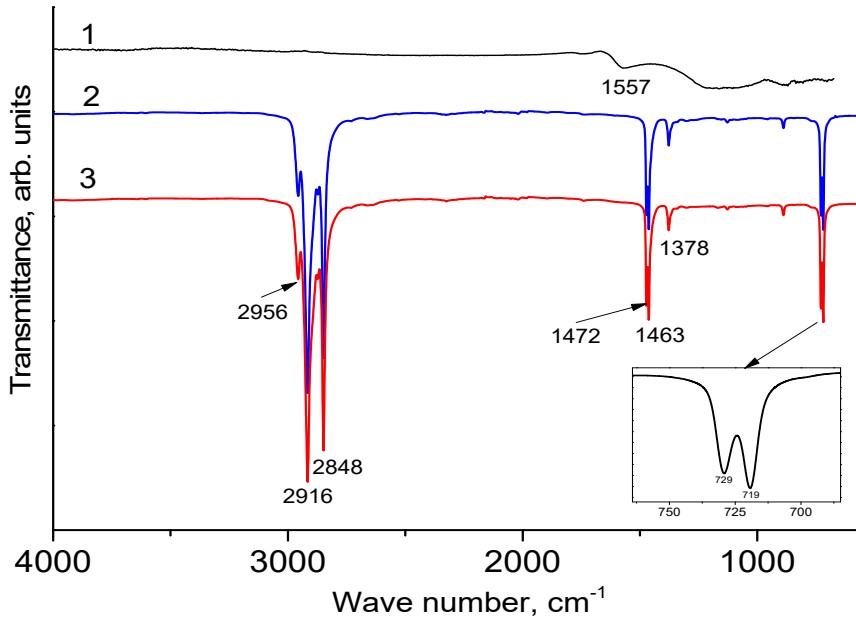

**Figure 2.** IR spectra of rGO (1), pure paraffin (2) and composite paraffin/rGO (3). Inset: IR spectrum of composite in the region of rocking vibrations of methylene groups.

On the IR spectrum of paraffin (Figure 2), eight peaks (2956, 2916, 2848, 1472, 1463, 1378, 729 and 719 cm$^{-1}$) can be distinguished. According to the literature data, the first three peaks were due to stretching vibrations of C–H bonds [27–29,36,37]. Peaks at 1472.7 and 1463.0 cm$^{-1}$ were due to bending vibrations of –CH$_2$ groups. The characteristic peak at 1378 cm$^{-1}$ was related to the C–H bending of the methyl groups [38,39]. Peaks at 729.2 and 719.3 cm$^{-1}$ refer to the rocking vibrations of methylene groups [38,40].

According to [41], the double absorption band in the region of the rocking vibrations of methylene groups was characteristic of the crystalline state of long-chain n-paraffins. The introduction of rGO did not significantly affect the IR spectrum. No noticeable shifts in the absorption bands in one direction or another were also recorded. In our opinion, the low sensitivity of the IR spectra to the presence of rGO in the composite was due to the low concentration of the additive (0.3 wt %). The IR spectra indicate that the combination between pure paraffin and rGO was only a physical connection.

### 3.3. Raman Spectra

Figure 3 shows the Raman spectra of paraffin and the composite in a range from 1000 to 1700 cm$^{-1}$. The Raman spectra of paraffins are well known [42–44]. The bands at 1070 and 1140 cm$^{-1}$ refer to the symmetric and antisymmetric vibrations of the C–C bonds. The band around 1300 cm$^{-1}$ is associated with twisting vibrations of CH$_2$ [43] groups. The three overlapping peaks in 1400–1500 cm$^{-1}$ region is a rather complex combination of CH$_2$ and CH$_3$ bending vibrations.

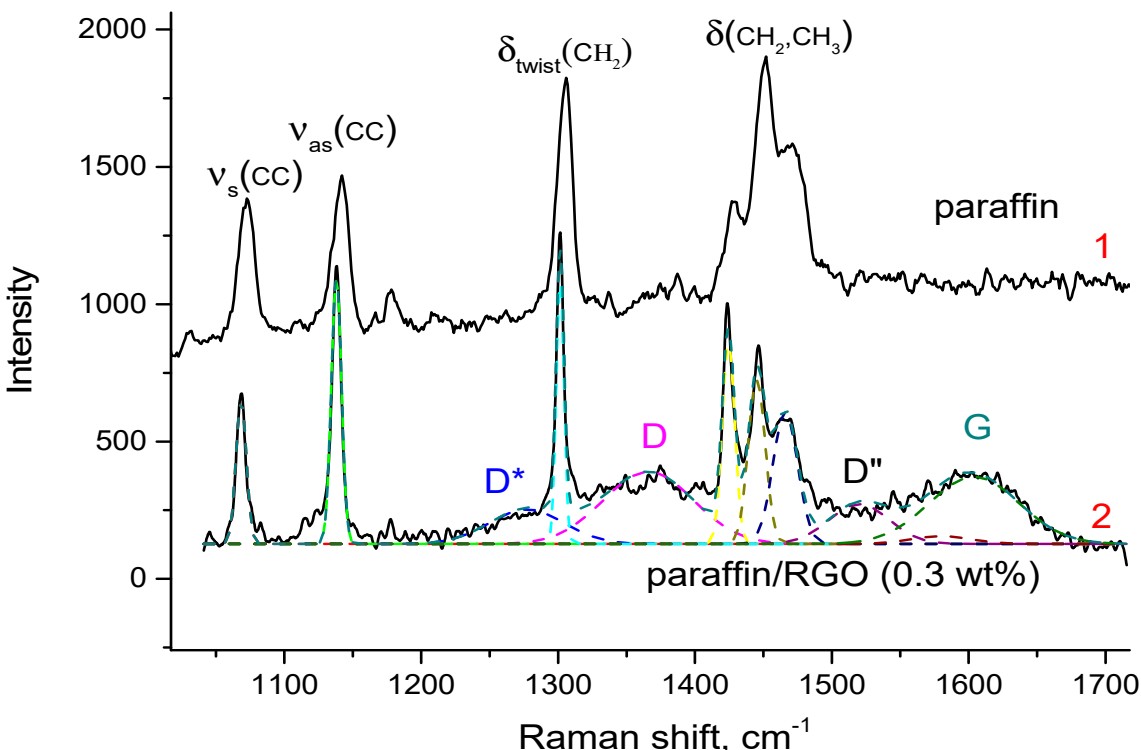

**Figure 3.** Raman spectra of pure paraffin (1) and composite paraffin/rGO (2).

It can be seen that the shape of the Raman spectra of the paraffin and the composite in the region of 1400–1500 cm$^{-1}$ were different. However, we did not associate these differences with the effect of rGO. The point is that the shape of the paraffin spectrum in this region depends on the temperature of the sample in the analysis zone. Since we did not control the temperature of the sample in the analysis zone, we can assume that the observed changes were due to different temperatures.

The introduction of rGO also led to the appearance in the Raman spectrum of a composite of bands characteristic of rGO (Figure 3). We note here that the paraffin-rGO system was previously studied by the Raman method (see, for example, [45,46]). In the Raman spectra of the composites given in these works, one can see only peaks characteristic of rGO. The absence in the Raman spectrum of the paraffin-rGO composite of peaks from paraffin, the proportion of which was 98%, was explained by the authors [45] by the fact that the paraffin was encapsulated by rGO sheets. In the present work, we obtained the Raman spectrum of the composite, where paraffin and rGO peaks were simultaneously present.

The assignment of individual peaks in the Raman spectrum of rGO is described (see, for example, [47,48]). In our study, we followed the authors of [49]. Table 1 lists the position, relative intensities, and half-widths of the individual peaks in the Raman spectrum of the composite, as well as similar parameters of the Raman spectrum from graphene oxide (for comparison). It can be seen from Table 1 that after the introduction of paraffin, the main peaks of the Raman spectrum of graphene oxide (D and G) narrow, and the $I_D/I_G$ ratio increase. We interpreted such changes in the Raman spectrum of rGO as an increase in the degree of its reduction. It is possible that rGO storage in the air led to water adsorption and partial oxidation. The treatment of rGO with warm liquid paraffin increased the degree of its reduction and prevented re-oxidation.

**Table 1.** The positions (*Pos*), full widths at half maximum (*FWHM*), and intensities (*Int*) of the peaks in the Raman spectra of the samples under study.

| Sample | Peak | *Pos*, cm$^{-1}$ | *FWHM*, cm$^{-1}$ | *Int*, % |
|---|---|---|---|---|
| | D* | 1114.2 | 156 | 6.2 |
| | D | 1348.5 | 192 | 38.9 |
| rGO | D″ | 1520.0 | 143 | 16.8 |
| | G | 1587.1 | 120 | 27.5 |
| | D′ | 1613.3 | 35 | 10.3 |
| | D* | 1144.2 | 64 | 5.2 |
| | D | 1365.9 | 78 | 60.1 |
| Paraffin/rGO | D″ | 1520.2 | 52 | 9.7 |
| | G | 1584.0 | 51 | 22.4 |
| | D′ | 1604.1 | 77 | 2.7 |

### 3.4. TG + DSC

Figure 4 shows the TG curves for paraffin and a composite based on it. The presented data indicate that the introduction of 0.3 wt. % rGO did not destroy but, on the contrary, slightly increased the thermal stability of paraffin. A similar effect in TG studies of paraffin and paraffin/rGO composites was observed [45,50]. Although only a physical interaction existed between the components of the composite, nevertheless, a small addition of rGO improved the thermal stability of the paraffin. It can be noted that weight loss began at temperatures above 140 °C when the paraffin was in a liquid state. The paraffin at these temperatures began to evaporate, and evaporation began with molecules with a lower molecular weight. A slight increase in thermal stability at the initial stage could be explained, for example, by the predominant adsorption of the most volatile paraffin molecules onto rGO sheets.

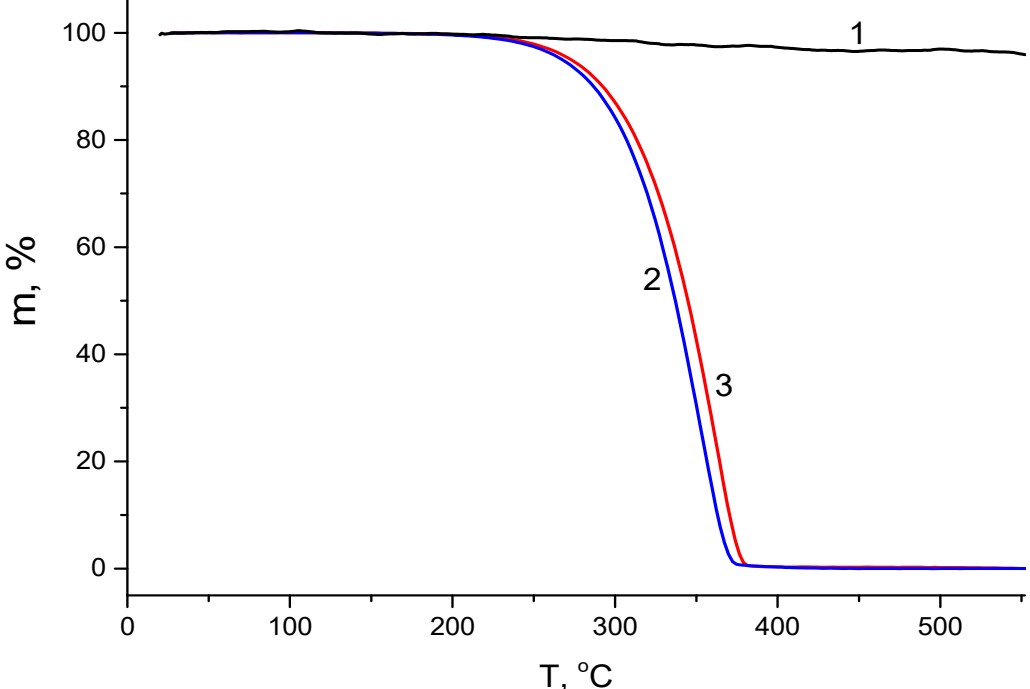

**Figure 4.** TG curves for rGO (1), pure paraffin (2) and composite paraffin/rGO (3).

It is well known that the thermophysical parameters of the solid–liquid transition and vice versa in the first heating cycle may differ from subsequent operating thermal cycles. For this reason, Figure 5 and Table 2 show the values that were obtained in the third thermal cycle. From the presented data, it can be seen that the melting temperature of the

composite decreased by only 0.11 °C compared to pure paraffin, while the solidification temperature increased by 0.71 °C. The introduction of rGO had a noticeable effect on the latent heat, and if the latent heat of melting increased only by 1.77 J/g, then the latent heat of solidification increased by 3.44 J/g.

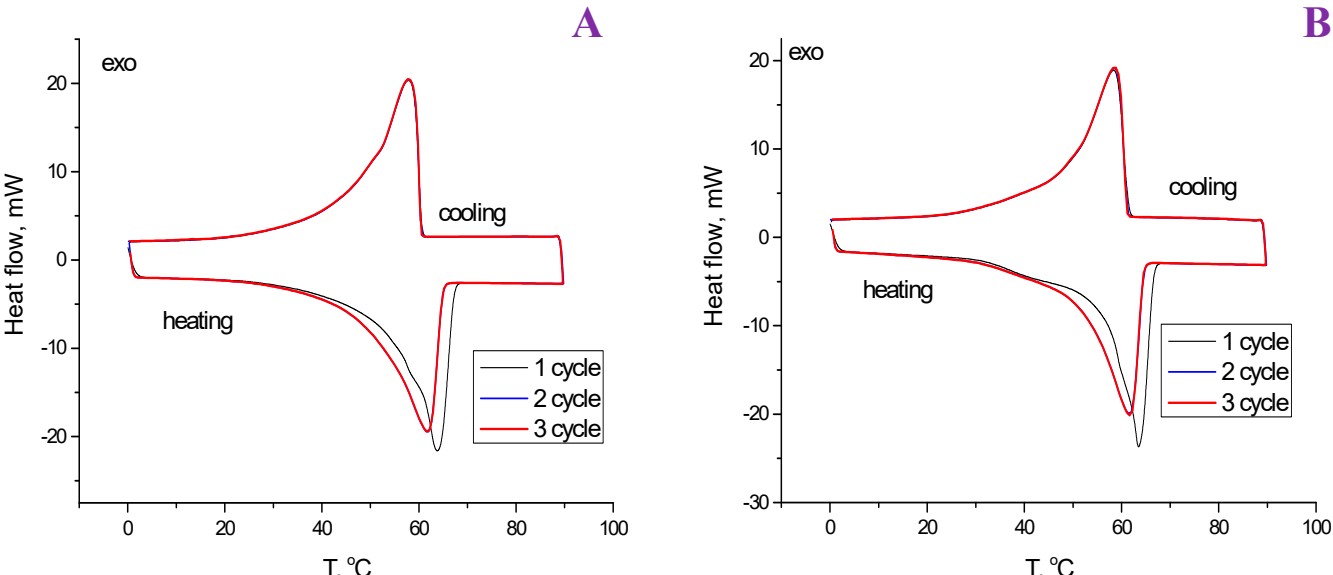

**Figure 5.** DSC curves for paraffin (**A**) and composite (**B**).

**Table 2.** Thermal properties of samples under study. (M–melting, S–solidification, $T^p$ – peak position on the DSC curve, ΔH is the enthalpy of transition). Dates were taken at 3-th cycle.

| Sample | $T^p_M$ (°C) | $\Delta H_M$ (J/g) | $T^p_S$ (°C) | $\Delta H_S$ (J/g) |
|---|---|---|---|---|
| Paraffin | 61, 83 | 182.31 | 57, 86 | 181.86 |
| Composite paraffin/rGO | 61, 72 | 184.08 | 58, 57 | 185.26 |

## 4. Discussion

### 4.1. About Burning Pure Paraffin and Composite

Candles of the same diameter and weight were made from paraffin and composite. It turned out that a small admixture of such a nanomaterial as reduced graphene oxide had a very strong effect on the candle-burning process (Figure 6). It could be seen that in the case of a black candle, the height of the flame was greater. When burning, the height of the black candle decreased faster than the height of the white candle. The reason for this was not only the slight reduction in the melting temperature but also the significantly improved thermal conductivity, which was obviously easy to understand.

### 4.2. About Heat Battery Charging

In Figure 7A, you can see photographs of two heat accumulators, the working fluid of which was pure paraffin and composite. Two-minute heating of the batteries by simultaneous immersion in boiling water was accompanied by an increase in their temperature to 51 °C (pure paraffin) and 53 °C (composite). These data were obtained using a thermocouple thermometer. These data were confirmed by a photograph taken with a thermal imager. It can be seen that the battery based on the composite was heated somewhat more efficiently. However, the difference was small.

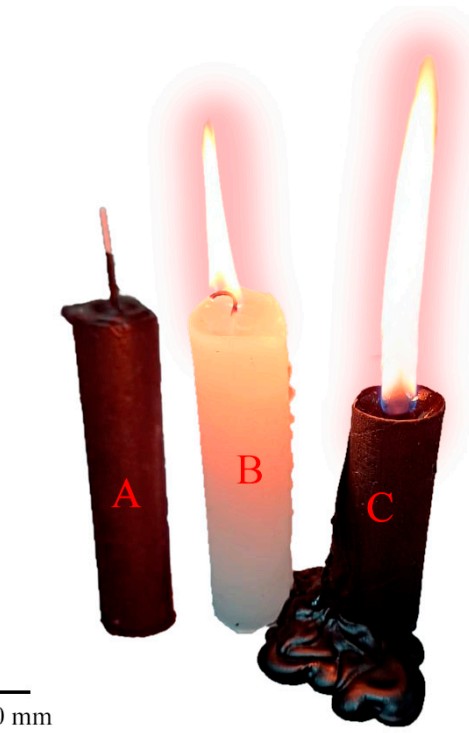

20 mm

**Figure 6.** Photo of candles which were made from paraffin/rGO composite (black candle **A**), paraffin (white burning candle **B**) and composite (black burning candle **C**). All candles initially had the same dimensions (diameter 27.6 mm, height 118.8 mm). During the burning time (10 min), the height of the white candle decreased by 9.7 mm, and that of the black candle by 38.4 mm.

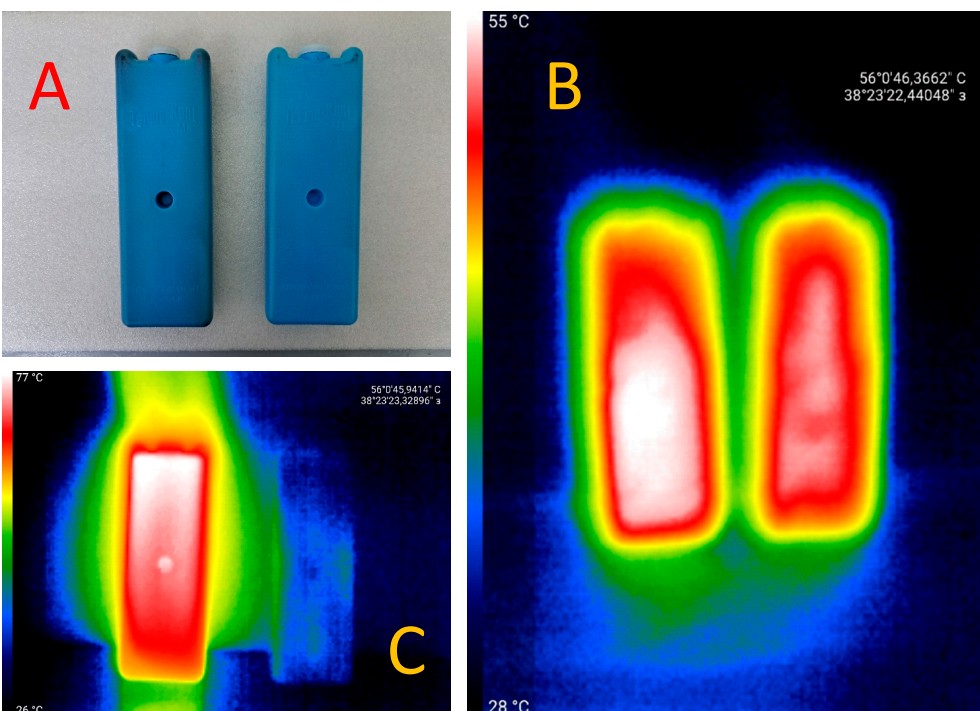

**Figure 7.** (**A**)-Optical photographs of batteries filled with pure paraffin (**right**) and composite (**left**); (**B**)-Photographs taken by a thermal imager of batteries filled with pure paraffin (on the right) and composite (on the left), after their 2 min stay in boiling water; (**C**)-Thermal imaging photographs of batteries filled with pure paraffin (**right**) and composite (**left**) after heating them for 2 min in a microwave oven (700 watts). The initial temperature of the batteries before the heating procedure was 23 °C. The weight of the working fluid (paraffin) in both batteries was 160 grams.

A large difference in the heating rate of the compared batteries occurred when using a microwave oven for heating. The two-minute heating of an accumulator based on pure paraffin led only to a small (up to 32 °C) increase in its temperature. During the same time, the temperature of the composite battery increased to 74 °C, which was higher than the phase transition temperature (see also Figure 7C)! This meant that 2 min of heating in a household microwave oven was enough to fully charge the battery.

The reason for this difference in the heating rate of the compared batteries in a microwave oven is due to the fact that reduced graphene oxide absorbs microwave radiation very efficiently. It is known from the literature that the heating of graphite oxide is accompanied by its exfoliation with the formation of MEGO [30]. The rapid heating of GO films in the microwave field can be accompanied by an explosion [31]. In the case of heating a battery based on a composite, the possibility of an explosion can be excluded since the total rGO concentration is very low, and the rGO particles used to create the composite also had small lateral dimensions.

## 5. Conclusions

A paraffin composite with reduced graphene oxide (0.3 wt %) was obtained at 70 °C by mechanical mixing followed by ultrasonic dispersion. The additive stained the paraffin black. The introduction of rGO had no significant effect on the IR spectrum of the paraffin; the peaks characteristic of rGO in the IR spectrum of the composite could not be identified. The introduction of this did not destroy but, on the contrary, slightly increased the thermal stability of the paraffin. A thorough study of paraffin and composite by DSC showed that the introduction of rGO increased the latent heat of fusion by 1.77 J/g and the latent heat of curing by 3.44 J/g. This seemed to be a small change compared to the initial values ($\Delta H_M \sim \Delta H_S \sim 180$ J/g).

Heat batteries made of pure paraffin and composite charge equally slowly when placed in boiling water. However, heating using microwaves proceeded differently for the compared batteries. For example, a two-minute heating of a battery based on pure paraffin led only to a slight (up to 32 °C) increase in its temperature. During the same time, the temperature of the battery based on the composite increased to 74 °C, which was higher than the phase transition temperature. The reason for this difference in the heating rate of the compared batteries in a microwave oven was due to the fact that the reduced graphene oxide absorbed microwave radiation very efficiently.

**Author Contributions:** Investigation, formal analysis, writing—original draft, S.A.B.; Investigation, formal analysis, Y.V.B., Investigation, visualization, formal analysis, writing—review and editing, E.N.K.; Investigation, formal analysis, E.V.D.; Investigation, formal analysis, V.N.V.; Review and editing, data curation, Z.L.; Review and editing, project administration, Y.M.S. All authors have read and agreed to the published version of the manuscript.

**Funding:** This work was supported by the Ministry of Science and Higher Education of the Russian Federation in the frame of state tasks (state registration Nos AAAA-A19-119032690060-9, 122040500074-1, and AAAA-A19-119061890019-5).

**Data Availability Statement:** Data supporting the reported results can be obtained on request from the authors.

**Acknowledgments:** The work was performed using the equipment of the Multi-User Analytical Center of IPCP RAS. This study was carried out with the use of resources of Competence Center of National Technology Initiative in IPCP RAS.

**Conflicts of Interest:** The authors declare no conflict of interest.

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
