# Peer review of "Fast Charging of a Thermal Accumulator Based on Paraffin with the Addition of 0.3 wt. % rGO"

_jcs, doi:10.3390/jcs7050193_

Round 1

Reviewer 1 Report

The authors prepared a paraffin/rGO (0.3 wt %) composite; and investigate the properties of the composite. However, there are some issues should be addressed before acceptance.

1.     Abstract. The novelty of this study should be highlighted in the abstract.

2.     Introduction. The introduction of rGO/GO is insufficient. Some papers including “Graphene Lubrication. https://doi.org/10.1016/j.apmt.2020.100662 ; Graphene superlubricity: A review. https://doi.org/10.1007/s40544-022-0681-y ; Quantum dots of graphene oxide as nano-additives trigger macroscale superlubricity with an extremely short running-in period. Materials Today Nano, 2022, 18: 100219, https://doi.org/10.1016/j.mtnano.2022.100219 ” should be cited.

3.     Figure 1. Scale bar should be added into the figure.

4.     Conclusion. The authors claimed that “The reason for this difference in the heating rate of compared batteries in a microwave oven is due to the fact that reduced graphene oxide absorbs microwave radiation very efficiently.” However, the absorbs microwave radiation was not even mentions in the results or discussion sections. Please improve.

Author Response

  1. Abstract. The novelty of this study should be highlighted in the abstract.

Authors reply:

The abstract has been rewritten.

Reviewer #1:

  1. Introduction. The introduction of rGO/GO is insufficient. Some papers including “Graphene Lubrication. https://doi.org/10.1016/j.apmt.2020.100662 ; Graphene superlubricity: A review. https://doi.org/10.1007/s40544-022-0681-y ; Quantum dots of graphene oxide as nano-additives trigger macroscale superlubricity with an extremely short running-in period. Materials Today Nano, 2022, 18: 100219, https://doi.org/10.1016/j.mtnano.2022.100219 ” should be cited.

Authors reply:

The references are cited.

 Reviewer #1:
3. Figure 1. Scale bar should be added into the figure.

 Authors reply:

In the new version of the manuscript a scale bar was added.

Reviewer #1:
4. Conclusion. The authors claimed that “The reason for this difference in the heating rate of compared batteries in a microwave oven is due to the fact that reduced graphene oxide absorbs microwave radiation very efficiently.” However, the absorbs microwave radiation was not even mentions in the results or discussion sections. Please improve.

Authors reply:

In the new version of the manuscript, "the absorbs microwave radiation" is discussed in the introduction.

Reviewer 2 Report

This study is interesting, and the results are well-discussed. My comments are as follows:

  1. It is necessary to give more important information in the abstract.
  2. The author needs to cut out some unimportant keywords.
  3. To better analyze the effect of the added rGO on the composite, it is necessary to show the IR spectra and TGA curve of rGO in Fig. 2 and Fig. 3, respectively.
  4. In the section “About heat battery charging", what is the influence of heating time for these two kinds of batteries when using a microwave oven for heating? This study only gives the data after heating for 2 minutes. After a long-time of heating, is it safe for this battery?

Reviewer 3 Report

In this manuscript entitled “Fast charging of a thermal accumulator based on paraffin with the addition of 0.3 wt. % rGO”, Baskakov and coauthors have reported on the investigation of rGO/paraffin composite. It is shown that the introduction of 0.3 wt. % rGO in paraffin stains it black. A thorough study of paraffin and related composite by DSC shows that the introduction of rGO increases the latent heat of fusion by 1.77 J/g, and the latent heat of curing by 3.44 J/g. Heating using microwaves proceeds differently for pure paraffin and rGO/paraffin batteries. The reason for this difference in the heating rate of compared batteries in a microwave oven is due to the fact that rGO absorbs microwave radiation very efficiently. Overall, this study is interesting. And the strategy is facile and cost-efficient. However, there are still issues to be addressed. Therefore, a major revision concerning the following comments are necessary.

1. More characterizations on the rGO/paraffin composite are needed. For example, SEM is recommended to be used to identify the distribution of rGO in paraffin.

2. As shown in Figure 3 (Page 5), the addition of rGO slightly increases the thermal stability of paraffin. What’s the underlying reason? It should be explained.

3. As this is a scientific paper, the description on the experimental phenomenon should be quantitative. Therefore, the burning time and the remaining height of the candle should be given in the description of Figure 5.

4. As shown in Figure 6 (Page 8), it seems that heat battery made of pure paraffin charges faster than heat battery made of composite when placed in boiling water. This indicates that the rGO/paraffin composite has a lower thermal conductivity, which is not consistent with previous studies. Why?

Round 2

Reviewer 1 Report

The manuscript was carefully revised by the authors, which can be accepted at present.

Reviewer 2 Report

The authors have made good revisions to the manuscript. However, there are still several minor errors, such as text format (see lines 125-130), unit superscript (see line 157), etc. The authors need to check the manuscript carefully before publication.

Reviewer 3 Report

The authors have addressed all my comments, and it is suggested that this manuscript is ready for publication.